# Genome-wide association analysis of Russian wheat aphid (*Diuraphis noxia*) resistance in *Dn4* derived wheat lines evaluated in South Africa

Lavinia Kisten[1,2]*, Vicki L. Tolmay[1]*, Isack Mathew[3], Scott L. Sydenham[4], Eduard Venter[2]

**1** Germplasm Development, ARC-Small Grain, Bethlehem, Free State, South Africa, **2** Department of Botany and Plant Biotechnology, University of Johannesburg, Johannesburg, Gauteng, South Africa, **3** School of Agricultural, Earth and Environmental Sciences, University of KwaZulu-Natal, Pietermaritzburg, KwaZulu-Natal, South Africa, **4** LongReach Plant Breeders Management Pty Ltd, York, Western Australia, Australia

* kistenl@arc.agric.za (LK); tolmayv@arc.agric.za (VLT)

**Data Availability Statement:** All relevant data files are available from the ArrayExpress database

## Abstract

Russian wheat aphid (RWA; *Diuraphis noxia* Kurdjumov) resistance on the 1D chromosome of wheat has been the subject of intensive research. Conversely, the deployment of the *Dn4* derived RWA resistant varieties diminished in recent years due to the overcoming of the resistance it imparts in the United States of America. However, this resistance has not been deployed in South Africa despite reports that *Dn4* containing genotypes exhibited varying levels of resistance against the South African RWA biotypes. It is possible that there may be certain genetic differences within breeding lines or cultivars that influence the expression of resistance. The aim of this study was to identify single nucleotide polymorphism (SNP) markers associated with resistance to South African RWA biotypes. A panel of thirty-two wheat lines were phenotyped for RWA resistance using four South African RWA biotypes and a total of 181 samples were genotyped using the Illumina 9K SNP wheat chip. A genome wide association study using 7598 polymorphic SNPs showed that the population was clustered into two distinct subpopulations. Twenty-seven marker trait associations (MTA) were identified with an average linkage disequilibrium of 0.38 at 10 Mbp. Four of these markers were highly significant and three correlated with previously reported quantitative trait loci linked to RWA resistance in wheat. Twenty putative genes were annotated using the IWGSC RefSeq, three of which are linked to plant defence responses. This study identified novel chromosomal regions that contribute to RWA resistance and contributes to unravelling the complex genetics that control RWA resistance in wheat.

## Introduction

The Russian wheat aphid (RWA; *Diuraphis noxia* Kurdjumov) is a significant pest of wheat globally. Since its introduction in South Africa in 1978 [1] yield losses of up to 60–90% [2]

([https://www.ebi.ac.uk/arrayexpress/](https://www.ebi.ac.uk/arrayexpress/)) under accession number E-MTAB-9671.

**Funding:** Funding for the research undertaken in this manuscript were sourced from the following funders: 1. South African Winter Cereal Trust [https://www.wintercerealtrust.co.za](https://www.wintercerealtrust.co.za) Grant WCT/W/2018/06 - Awarded to VT PhD bursary - Awarded to LK 2. National Research Foundation - Research and Technology Fund [https://www.nrf.ac.za](https://www.nrf.ac.za) Grant RTF1505 2911 8466 - Awarded to VT 3. National Research Foundation - Professional Development Programme [https://www.nrf.ac.za](https://www.nrf.ac.za) Block Grant PDP160318160924 - Awarded to VT via Jasper Rees PhD bursary - Awarded to LK 4. LongReach Plant Breeders Management Pty Ltd [https://www.longreachpb.com.au](https://www.longreachpb.com.au) Salary of SLS The funders [1-3] mentioned above had no role in study design, data collection and analysis, decision to publish, or preparation of the manuscript. The funder (LongReach Plant Breeders Management Pty Ltd) provided support in the form of salaries for author SLS but did not have any additional role in the study design, data collection and analysis, decision to publish, or preparation of the manuscript. The specific roles of these authors are articulated in the 'author contributions' section.

**Competing interests:** Scott L. Sydenham is presently an employee of LongReach Plant Breeders Management Pty Ltd. This does not alter our adherence to PLOS ONE policies on sharing data and materials.

have been incurred due to the pest. Although, an ubiquitous pest, infestation rates are low with sporadic occurrences in areas limited to the central interior and the Western Cape of the country. In order to negate the effect of this pest host plant resistance is considered the most efficient and sustainable approach to manage infestations. In South Africa, an integrated RWA management approach was attained with resistant cultivars developed through conventional breeding methods. To date, forty three cultivars with resistance to RWA have been released [3]. However, conventional breeding is a long process and the development and release of stable cultivars can take up to twelve years [4]. This process has now become too slow to address challenges presented by the development of new aphid biotypes. There are currently five RWA biotypes in South Africa with the newly observed biotype, RWASA5, identified in 2019 [5]. In the United States of America (USA), eight RWA biotypes have been identified [6]. Currently, efforts are being made to expedite the breeding process through marker-assisted selection, which can significantly reduce the number of breeding cycles required to develop stable cultivars.

Developing pest resistant cultivars requires an understanding of complex genetic components involved in pest resistance as well as the identification of sources of resistance that can be incorporated into breeding programs. To date, fifteen RWA resistance genes have been reported. Various studies have mapped *Dn1*, *Dn2*, *Dn5* [7], *Dn6* [8], *Dn8* [9], *Dnx* [10], *Dn2401* [11], *Dn626580* [12], *Dn100695* [13] and *Dn10* [14] to the 7D chromosome of wheat. The recessive resistance gene, *dn3*, resistance gene was derived from line SQ24 of *Aegilops tauschii* and is currently unmapped [15]. Reports indicate that the resistance genes *Dn4* and *Dn9* are located on the short and long arm of the 1D chromosome, respectively [9, 16, 17]. *Dn7*, a gene derived from a rye accession, was transferred to the 1RS/1BL translocation in wheat [18, 19]. Presently the chromosomal location of *Dny* is unknown but is assumed to be located on chromosome 1D [20]. Many studies allude to the existence of multiple alleles or clusters of genes [3] on both 1D and 7D. Currently, none of the RWA resistance genes have been cloned or fully characterised, hence the location of these genes was inferred through markers in genetic mapping studies. Genetic components involved in RWA resistance have also been reported on chromosomes 1A, 1B, 1D, 3A, 3B, 4A, 4D, 5B, 5D, 6A, 6D, 7A, 7B and 7D [7, 9, 16, 18, 19, 21–36].

The *Dn4* resistance gene was previously reported as a single dominant gene [16]. It was the first source of RWA resistance identified in the USA and provided resistance against the first USA biotype RWA1 [16]. However, upon the identification of RWA2, the resistance conferred by the *Dn4* gene was no longer considered to be effective [37]. Although, the study by Puterka et al. [38] confirmed that whilst *Dn4* is no longer effective against RWA2, it was found to confer resistance to RWA6. In South Africa, the biotype RWASA3 was observed to be virulent to *Dn4* in some accessions [39]. However, to our knowledge *Dn4* has never been commercially deployed in South Africa.

Despite previous reports indicating that the majority of RWA resistance function in a qualitative relationship, several recent studies have mapped quantitative trait loci (QTL) for resistance against various RWA biotypes [32, 33, 40]. Furthermore, a study conducted by Tolmay et al. [39] revealed that genotypes containing the same donor line, PI372129 (*Dn4* donor), differed in response to RWASA3 with the majority displaying a resistant phenotype. This suggests that not all cultivars and breeding lines with resistance derived from PI372129 are equally resistant. These findings imply that there may be certain genetic differences within breeding lines or cultivars that influence the expression of resistance.

The objective of this study was to use a genome wide association study (GWAS) on a set of *Dn4* containing resistant and susceptible lines to identify single nucleotide polymorphism (SNP) markers associated with resistance to four of the South African RWA biotypes. Several

association studies of RWA resistance have been conducted on barley [41, 42], however there is only one GWAS investigating RWA resistance in wheat [43]. This study is unique as it evaluates resistance to the South African complex of RWA biotypes. Information generated from this study will increase our knowledge on the genetic control of the RWA and contribute to breeding for resistance to RWA.

## Material and methods

### Preliminary work

A panel of wheat genotypes containing the RWA resistance genes *Dn4* or *Dny* (S1 Table) were evaluated against the RWASA3 and thereafter screened with fourteen SSR (simple sequence repeats) markers located within and around the *Dn4* locus (S2 Table). All SSR marker primer pairs were synthesised by Integrated DNA Technologies. The fourteen SSR markers consist of the two initially published *Dn4* associated markers, *Xgwm106* and *Xgwm337* [26] and twelve additional SSR markers which were selected from chromosome 1D of the wheat consensus map [44]. Each specific BARC, CFD, WMC and GWM SSR marker primer pair sequence and annealing temperature (S2 Table) were obtained from the wheat molecular marker database on GrainGenes 2.0 [45]. Only polymorphic markers were used to validate allele variation observed.

PCR reactions were performed in a 20μl volume with 150ng of DNA template using the KAPA 2X Ready Mix PCR Kit (KAPA Biosystems, Cape Town, South Africa, currently owned by Roche, B, Switzerland) according to the manufacturer's recommendations. PCR reactions were performed in a MyCycler™ Thermal Cycler (Bio-Rad Laboratories, Inc, Johannesburg, South Africa) programmed with the following parameters: 5 min at 95 ˚C followed by 35 cycles of 30 s at 95 ˚C, 30 s of annealing at respective temperatures for each primer pair (S2 Table), 30 s at 72 ˚C and a final extension step of 5 min at 72 ˚C. PCR amplicons were separated on 3–4% (w/v) Certified Low Range Ultra Agarose high-resolution gels (Bio-Rad Laboratories, Inc, USA) stained with SYBR® safe (Invitrogen products supplied by Thermo Fisher Scientific Pty Ltd, Germiston, South Africa) gel stain and were electrophoresed at 5 V/cm for two hours. Agarose gels were digitally photographed under UV light using the Bio-Rad Molecular Imager Gel Doc™ XR. The PCR amplicons were sized visually and with image Lab™ gel analysis software. The results obtained from the SSR marker screening formed the basis for conducting the association analysis.

### Genome wide association study

**Plant material.**   This study evaluated a total of 181 individuals developed from a panel of thirty-two wheat genotypes. This consisted of twenty-four lines reported to contain the RWA resistance gene *Dn4*, two lines containing the *Dny* gene and one line containing a combination of *Dn4* and *Dnx* and five control lines (Table 1). The twenty-seven lines containing RWA resistance genes comprised of ten winter-type American cultivars, thirteen spring-type breeding lines developed in Montana, USA, three winter-type breeding lines obtained from the International Winter Wheat Improvement Programme (iwwip.org) based in Turkey and the *Dn4* donor line PI372129. Gariep, Yumar and Pan 3144 were used as differential checks for the RWA biotypes while Hugenoot and CItr2401 were used as the susceptible and resistant checks respectively. The test genotypes were developed using the American biotype RWA1. In order to prevent the possibility of unnecessary complexity, single plants of each line from a mother plant with a known RWA damage score to the South African biotype RWASA2 were used for downstream analyses.

**Table 1. Panel of genotypes used in the evaluation of *D. noxia* resistance with four South African biotypes.**

| Genotypes | Resistance gene(s) | Growth habit | RWASA2 damage score[1] | Pedigree[2] |
|---|---|---|---|---|
| Ankor | *Dn4* | Winter | 4 | Akron/Halt//4*Akron |
| BondCL | *Dn4* | Winter | 3 | Yumar//TXGH12588-120*4/FS-2 |
| Corwa | *Dn4* | Winter | 7 | Sumner/CO820026//PI372129/3/TAM107 |
| Halt | *Dn4* | Winter | 5 | Sumner/CO820026//PI372129/3/TAM107 |
| Hatcher | *Dn4* | Winter | 3 | Yuma/PI372129//TAM200/3/4*Yuma/4/KS91H184/Vista |
| Outlook | *Dn4* | Winter | - | PI372129/2*Amidon//MT-7810/MT-7926 |
| Prowers99 | *Dn4* | Winter | 7 | CO850060/PI372129//5*Lamar |
| Ripper | *Dn4, Dnx* | Winter | 4 | I220127/P5//TAM200/KS87H66(CO940606)/CO850034/PI372129//5*TAM107(TAM107R-2) |
| Stanton | *Dny* | Winter | 3 | PI220350/KS87H57//TAM200/KS87H66/3/KS87H325 |
| ThunderCL | *Dn4* | Winter | 6 | KS01-5539/CO99W165(FS2/KS97HW150//KS97HW349/3/KS92WGRC25/Halt) |
| PI327129 | *Dn4* | Winter | - | Turcikum57, Landrace donor of *Dn4* from Turkmenistan |
| MTRWA92-91 | *Dn4* | Spring | 5 | PI372129/*2Pondera |
| MTRWA92-93 | *Dn4* | Spring | 2 | PI372129/*2Pondera |
| MTRWA92-114 | *Dn4* | Spring | 3 | PI372129/*2Pondera |
| MTRWA92-115 | *Dn4* | Spring | 3 | PI372129/*2Pondera |
| MTRWA92-120 | *Dn4* | Spring | 4 | PI372129/*2Pondera |
| MTRWA92-121 | *Dn4* | Spring | 6 | PI372129/*2Pondera |
| MTRWA92-123 | *Dn4* | Spring | 3 | PI372129/*2Pondera |
| MTRWA92-145 | *Dn4* | Spring | 4 | PI372129/*2Newana |
| MTRWA92-149 | *Dn4* | Spring | 4 | PI372129/*2Newana |
| MTRWA92-150 | *Dn4* | Spring | 3 | PI372129/*2Newana |
| MTRWA92-155 | *Dn4* | Spring | - | PI372129/*2Newana |
| MTRWA92-158 | *Dn4* | Spring | 3 | PI372129/*2Newana |
| MTRWA92-161 | *Dn4* | Spring | - | PI372129/*2Newana |
| 18FAWWON-SA 57 | *Dny* | Winter | - | KS99-5-16(94HW98/91H153)//Stanton/KS98HW423(JAG/93HW242) |
| 18FAWWON-SA 62 | *Dn4* | Winter | - | CO970547/Prowers99 |
| 18FAWWON-SA 64 | *Dn4* | Winter | - | CO980862/Lakin |
| Hugenoot | Susceptible check | Winter | S | Betta//Flamink/Amigo |
| Gariep | RWASA2, Differential check | Winter | S | SA-1684/4*Molopo |
| Yumar | RWASA3, Differential check | Winter | R | Yuma/PI372129//CO-850034/3/4*Yuma |
| Pan3144 | RWASA4, Differential check | Winter | R | Not available |
| CItr2401 | Resistant check | Winter | R | PI9781, Landrace from Tajikistan |

[1] RWASA2 damage scores of parental plants of lines used in this study that was obtained from prior screening analysis.

[2] Pedigree information obtained from the Genetic Resources Information System for Wheat and Triticale (GRIS) database (http://wheatpedigree.net/).

**Phenotyping.** Phenotypic evaluations were conducted in a glasshouse facility of the Agricultural Research Council-Small Grain, Bethlehem, the Free State province, South Africa (28° 09′55.12″S, 28°18′32.97″E). The thirty-two test genotypes were screened with four South African biotypes (RWASA1–4) using a 21-day seedling assay [39] in a split plot design with three replicates. For each replicate, five seeds of each genotype were planted in individual cones for

inoculation with each of the four biotypes. The cones were randomly assigned a plot position in each of the three replicates and infested at the two-leaf growth stage with five mixed-instar aphids of the relevant biotype. Twenty one days post infestation individual plants were scored using a 1–10 damage rating scale where 1 is highly resistant and 10 is highly susceptible [39].

Phenotypic data were subjected to analysis of variance (ANOVA) using GenStat® 18th edition [46] after testing for normality. The linear mixed model was used with genotype as a random factor while the biotype was a fixed factor. Subsequently, a best test analysis was conducted using the software SAS 9.2 [47] at a confidence of 95% to rank genotypes from the most resistant to the most susceptible for each biotype [48]. The genotypes were further subdivided into three categories, resistant (damage rating of 1–4), susceptible (damage rating of 7–10) and an intermediate category (damage rating of 5–6).

**Plant sampling and DNA isolation.**   Immediately after scoring, leaf tissue of test plants was sampled according to the bulked sample analysis method [49], where the five most resistant plants and the five most susceptible plants were bulked per line per biotype. The test individuals were chosen based on the phenotypic response of the thirty-two genotypes to four RWA biotypes. After inoculation and disease severity scoring, plants of the same genotype in the same replicate that exhibited differential response to RWA were bulked separately. A genotype could potentially have individual plants that exhibited susceptibility, intermediate and resistance to RWA. The individuals in the intermediate reaction grouped were not sampled. Only individuals with extreme i.e. either susceptible or resistant, reactions were sampled and bulked accordingly per biotype. Some genotypes consisted of individuals with similar reaction to a particular RWA biotype, in which case a single bulk sampled would be collected. Other genotypes consisted of individuals with differential reaction and at most, two bulk samples of susceptible and resistant samples would be collected per biotype. In total, 181 bulks, composed of like individuals with differential reaction to the four RWA biotypes were sampled for genotyping.

Genomic DNA was isolated from the bulked samples using a modified cetyltrimethylammonium bromide (CTAB, Sigma-Aldrich, Sandton, South Africa) extraction protocol [50]. The DNA concentration and quality were quantified using a Nanodrop 2000 Spectrophotometer (Thermo Scientific Pty Ltd, USA). Prior to downstream genotyping applications DNA samples were diluted to 50 ng/μl with 1x Tris-EDTA buffer. A total of 181 bulked samples were obtained that were used for subsequent SNP genotyping.

**Genotyping.**   Whole-genome screening of the 181 bulked samples was conducted at the Agricultural Research Council–Biotechnology Platform (Pretoria, South Africa) using the 9K Illumina Infinium iSelect BeadChip (Illumina Inc., San Diego, USA) [51]. A genotypic panel of 8632 SNP markers distributed on 21 wheat chromosomes was used in this study. Physical positions of the SNP markers were identified by aligning the markers to the reference genome of Chinese Spring (International Wheat Genome Sequencing Consortium (IWGSC), RefSeq v1.1) [52]. The SNP markers and individuals with more than 10% missing data and SNP alleles with less than 5% minor allele frequency were pruned using PLINK v.1.07 [53, 54] and were not considered for further analysis.

Following quality control, the population structure of the trimmed sample set that included 169 bulked samples was inferred using the Bayesian clustering algorithm implemented in STRUCTURE v.2.3.4 [55] based on 7598 polymorphic SNP markers distributed across the wheat genome. An admixture model with 10 000 burn-in and 10 000 Markov Chain Monte Carlo (MCMC) cycles was used. A series of K-values from 1–10 were tested in 20 independent runs. STRUCTURE HARVETSER Web v.0.6.94 [56] was used to determine the optimal number of population clusters and sub-clusters based on the Evanno method [57]. The GWAS was conducted using the GAPIT program [58] in the R software v.3.4.2 environment [59] using a

compressed mixed linear model (CMLM), which takes into consideration both the population structure and kinship matrix. The population structure was treated as a fixed factor whilst the kinship matrix was treated as a random factor. The Bonferroni correction method is conservative and known for producing a high rate of false negatives. Therefore, in addition to the Bonferroni method, a false discovery rate (FDR) adjusted p-value < 0.0001 was also considered as a threshold for significant markers.

Linkage disequilibrium (LD) was estimated as the frequency of the squared allele correlations ($R^2$) for markers with known positions using the GAPIT program [58] in R v.3.4.2 [59] at a significance level of p-values < 0.0001. LD decay was observed by plotting the $R^2$ values against the physical distance, in mega base pairs (Mbp) and a smoothing line was fitted to the data. The LD decay was estimated at the point where the curve exhibits the highest decay. Additionally, LD was calculated between SNP markers with significant associations with the phenotype and was graphically presented using the LDHeatmap package [60] in R v.3.4.2 [59].

The genetic sequences of significant markers were analysed using BLASTn™ against the IWGSC RefSeq v1.1 [7] database to identify potential candidate genes. The position of the candidate genes on the wheat chromosomes was depicted using KnetMiner [14]. Molecular functions and annotations of the candidate genes were sourced through the UniProt database [15] and available literature.

## Results

### Preliminary work

Eight of the 14 SSR markers tested (*Xwmc222*, *Xwmc429*, *Xwmc489*, *Xcfd21*, *Xcfd59*, *Xcfd65*, *Xcfd72* and *Xbarc99*) were not significantly polymorphic within or between *Dn4* and *Dny* containing germplasm. The remaining six markers (43%) were polymorphic between and within different *Dn4* and *Dny* genotypes, of which five markers (36%) (*Xgwm337*, *Xgwm191*, *Xgwm106*, *Xbarc119* and *Xcfd92*) were highly polymorphic. Marker, *Xcfd61* (7%) was not polymorphic within *Dn4* or *Dny* genotypes. However, *Xcfd61* was exclusively polymorphic between *Dn4* and *Dny* containing genotypes (Table 2).

Three different alleles were amplified for marker *Xgwm337* across all genotypes (Table 2). The observed alleles were at 175 bp, 195 bp and 225 bp in the test genotypes. The 175 bp fragment present in PI372129 (*Dn4*), was also observed in Halt, Corwa, Thunder and all the MTRWA92 lines. The 195 bp fragment was present in susceptible checks Hugenoot and Yuma as well as in *Dn4* carrying genotypes Hatcher, Prowers99, BondCL, ThunderCL and Yumar. In Corwa, which is reported to be a sister line of Halt, the 175 bp fragment and the 225 bp fragment were amplified in different single plants. The same 225 bp fragment was also observed in cultivars Ankor, Ripper and ThunderCL. ThunderCL however, showed mixed alleles, with the 195 bp fragment present in some individual plants. Within the 18 FAWWON-SA lines 62 (plants 62–1 to 62–5) and 64 (plants 64–1 to 64–3) individual plants showed amplification of different fragments for GWM337. Plants 62–1, 62–2 and 62–3 of 18 FAWWON-SA 62 line with Prowers99 and Halt in its pedigree, had the 195 bp fragment present, while plants 62–4 and 62–5 contained the 175 bp fragment. The same 175 bp fragment was present in all three plants of the 18 FAWWON-SA 64 line. Two of the MTRWA92 breeding lines (91 and 120), contained the 195 bp fragment, while the remaining eleven lines (93, 114, 115, 121, 123, 145, 149, 150, 155, 158 and 160) had the 175 bp fragment. In PI586956 two fragments, 195 bp and 225 bp, were amplified and in the *Dny* carrying cultivar Stanton, only the 195 bp fragment was amplified. The 225 bp fragment amplified in all plants of 18 FAWWON-SA 57 line.

Three different alleles amplified for marker *Xgwm106* with a 125 bp, 140 bp and a null allele observed (Table 2). The 125 bp fragment was amplified in PI372129, Halt, BondCL, Corwa,

**Table 2. Allelic variation observed for markers *Xgwm337*, *Xgwm106*, *Xcfd92* and *Xcfd61*.**

| Genotype | Gene | RWASA3 Score | *Xgwm337* | | | *Xgwm106* | | | *Xcfd92* | | *Xcfd61* | |
|---|---|---|---|---|---|---|---|---|---|---|---|---|
| | | Mean DR | 175[1] | 195[1] | 225[1] | null | 125[1] | 140[1] | null | 250[1] | 190[1] | 270[1] |
| Ankor | Dn4 | 7.2 | 0% | 0% | 100% | 100% | 0% | 0% | 0% | 100% | 100% | 0% |
| Bond CL | Dn4 | 7.3 | 0% | 100% | 0% | 0% | 100% | 0% | 0% | 100% | 100% | 0% |
| Corwa | Dn4 | 8.3 | 25% | 0% | 75% | 0% | 100% | 0% | 25% | 75% | 100% | 0% |
| Halt | Dn4 | 7.2 | 100% | 0% | 0% | 0% | 100% | 0% | 100% | 0% | 100% | 0% |
| Hatcher | Dn4 | 7.8 | 0% | 100% | 0% | 0% | 25% | 75% | 82% | 18% | 100% | 0% |
| Prowers99 | Dn4 | 7.8 | 0% | 100% | 0% | 100% | 0% | 0% | 0% | 100% | 100% | 0% |
| Ripper | Dn4 | 5.7 | 0% | 0% | 100% | 20% | 80% | 0% | 89% | 11% | 100% | 0% |
| Stanton | Dny | 8.7 | 0% | 100% | 0% | 100% | 0% | 0% | 0% | 100% | 0% | 100% |
| ThunderCL | Dn4 | 8.7 | 36% | 66% | 1% | 100% | 0% | 0% | 0% | 100% | 100% | 0% |
| Yumar | Dn4 | 7.8 | 0% | 100% | 0% | 0% | 100% | 0% | 30% | 70% | 100% | 0% |
| PI372129 | Dn4 | 6.3 | 100% | 0% | 0% | 0% | 100% | 0% | 100% | 0% | 100% | 0% |
| PI586956 | Dny | 8.2 | 0% | 20% | 80% | 100% | 0% | 0% | 0% | 100% | 0% | 100% |
| MTRWA92-91 | Dn4 | 5.5 | 75% | 25% | 0% | 100% | 0% | 0% | 0% | 100% | 100% | 0% |
| MTRWA92-93 | Dn4 | 5.8 | 100% | 0% | 0% | 100% | 0% | 0% | 100% | 0% | 100% | 0% |
| MTRWA92-114 | Dn4 | 5.5 | 100% | 0% | 0% | 100% | 0% | 0% | 100% | 0% | 100% | 0% |
| MTRWA92-115 | Dn4 | 5.7 | 100% | 0% | 0% | 100% | 0% | 0% | 100% | 0% | 100% | 0% |
| MTRWA92-120 | Dn4 | 5.7 | 75% | 25% | 0% | 100% | 0% | 0% | 0% | 100% | 100% | 0% |
| MTRWA92-121 | Dn4 | 5.9 | 100% | 0% | 0% | 100% | 0% | 0% | 100% | 0% | 100% | 0% |
| MTRWA92-123 | Dn4 | 5.3 | 100% | 0% | 0% | 100% | 0% | 0% | 100% | 0% | 100% | 0% |
| MTRWA92-145 | Dn4 | 6.8 | 100% | 0% | 0% | 100% | 0% | 0% | 0% | 100% | 100% | 0% |
| MTRWA92-149 | Dn4 | 5.7 | 100% | 0% | 0% | 100% | 0% | 0% | 100% | 0% | 100% | 0% |
| MTRWA92-150 | Dn4 | 6.2 | 100% | 0% | 0% | - | - | - | 100% | 0% | 100% | 0% |
| MTRWA92-155 | Dn4 | 6.2 | 100% | 0% | 0% | 100% | 0% | 0% | 100% | 0% | 100% | 0% |
| MTRWA92-158 | Dn4 | 5.5 | 100% | 0% | 0% | 100% | 0% | 0% | 100% | 0% | 100% | 0% |
| MTRWA92-160 | Dn4 | 6.2 | 100% | 0% | 0% | 100% | 0% | 0% | 0% | 100% | 100% | 0% |
| 18 FAWWON-SA 57 | Dny | 9.0 | 0% | 0% | 100% | 100% | 0% | 0% | 0% | 100% | 0% | 100% |
| 18 FAWWON-SA 62 | Dn4 | 7.6 | 18% | 82% | 0% | 100% | 0% | 0% | 40% | 60% | 100% | 0% |
| 18 FAWWON-SA 64 | Dn4 | 7.8 | 100% | 0% | 0% | 0% | 100% | 0% | 83% | 17% | 100% | 0% |
| Yuma | S check | 9.0 | 0% | 100% | 0% | 100% | 0% | 0% | 0% | 100% | - | - |
| Hugenoot | S check | 9.0 | 0% | 100% | 0% | 100% | 0% | 0% | 0% | 100% | - | - |
| CItr2401 | R check | 4.4 | - | - | - | 0% | 0% | 100% | 100% | 0% | - | - |

DR = damage rating.

R check = resistant check.

S check = susceptible check.

[1] Alleles are in base pairs.

Yumar and all of the 18 FAWWON-SA 64 plants. All MTRWA92 lines except 150 had a null allele. The null allele was also observed in Ankor, Prowers99, ThunderCL, in all 18 FAW-WON-SA 57 and 62 plants, PI586956, Stanton and in the susceptible checks Yuma and Hugenoot. In Cltr2401 the 140 bp fragment was amplified, while in Hatcher both 125 and 140 bp fragments occurred in individual plants. Ripper contained both the null allele and the 125 bp fragment.

Marker *Xcfd92* amplified two distinct alleles that consisted of a 250 bp fragment and a null allele (Table 2). The null allele was observed in PI372129, Cltr2401, Halt, 18 FAWWON-SA plants 62–4, 62–5, 64–1, 64–2, 64–3, and MTRWA92 lines 93, 114, 115, 121, 123, 149, 150, 155

and 158. The 250 bp fragment amplified in susceptible Yuma and Hugenoot, and the resistant cultivars Ankor, Bond, 18 FAWWON-SA plants 62–1, 62–2 and 62–3, MTRWA92 lines: 91, 120, 145 and 160, Prowers99, Thunder and Yumar. Genotypes Hatcher, Ripper, Yumar and Corwa had different fragments in individual plants, some containing the null allele and others the 250 bp fragment (Table 2). *Dny* containing genotypes, all 18 FAWWON-SA 57 plants, PI586956 and Stanton amplified the 250 bp fragment.

Marker *Xcfd61* amplified two different alleles at 190 bp and 270 bp. All *Dn4* containing genotypes had the 190 bp allele present and the 270 bp fragment was specific to all *Dny* containing genotypes namely, 18 FAWWON-SA 57, Stanton and PI586956.

The allelic variation observed across the different SSR markers located in and around the documented *Dn4* region on the 1D chromosome were associated with or partially explained the varying levels of resistance when screened with RWASA3.

## Genome wide association study

**Phenotypic evaluation.** Significant variation in the RWA damage rating means of the four biotypes was observed across the study panel (S3 Table). Phenotypic variation was also recorded among replications, despite the use of seeds from a single mother plant. Additionally, the phenotype varied through generations. The RWASA2 damage means for the majority of the lines in the test panel did not correspond with its respective mother plant RWA damage score (S3 Table and Table 1).

Wide-ranging infestation responses were observed for each biotype, from resistant to moderate resistance or susceptible. Control lines notwithstanding, the damage means for RWASA1, RWASA2, RWASA3 and RWASA4 ranged from 4 to 7; 4 to 8; 5 to 9 and 6 to 9, respectively (S3 Table). RWASA1 was noted as the least virulent biotype and RWASA4 was the most virulent on this panel of germplasm.

The genotypes ranked according to resistance levels are shown in Table 3. Only two of the 32 genotypes tested showed resistance to all four biotypes, namely, the resistant check CItr2401 and the original *Dn4* cultivar, Halt. Genotypes MTRWA92-161 and MTRWA92-93 were moderately resistant to RWASA2 and showed a high level of resistance to RWASA1, RWASA3 and RWASA4. Additionally, MTRWA92-155 showed a high level of resistance to RWASA1-3 and was moderately resistant to RWASA4. The susceptible check, Hugenoot, was the only genotype susceptible to all four biotypes.

**Population structure.** The population structure analysis delineated the panel into two distinct clusters (Fig 1A). Cluster 1 comprised 102 bulked samples of winter-type cultivars and breeding lines from the United States and Turkey while Cluster 2 contained 67 bulked samples of spring-type breeding lines that were developed in Montana in the United States (Fig 1A). This was further substantiated by the kinship matrix that showed a clear grouping of the genotypes into two main clusters and separate sub-clusters (Fig 1B). Cluster 2 exhibited the highest heterozygosity with an average of 0.21 while cluster 1 showed an average of 0.17. The mean fixation index ($F_{st}$) for clusters 1 and 2 were 0.62 and 0.31, respectively.

**Marker trait associations.** Russian wheat aphid resistance among the four biotypes was subjected to GWAS using 7 598 polymorphic SNP markers. A total of 27 marker trait associations (MTA) were identified to be in significant (FDR < 0.05; p < 0.0001) association with RWA resistance. The markers were mapped on 12 of the 21 wheat chromosomes (Fig 2, Table 4). Chromosomes with high numbers of MTA identified include: 6B with eight MTA, 1B with four MTA and 6A with three MTA (Fig 2; Table 4). A single MTA was identified on chromosome 1D (Tables 4 and 5). Discounting markers presumed to co-localise with reported QTL markers, 24 novel MTA were detected (Table 5). Four MTA were identified with the

**Table 3. Resistance grouping and ranking of genotypes using four South African Russian wheat aphid biotypes.** Genotypes were ranked based on their resistance level to the respective biotypes.

| Resistance | Russian Wheat Aphid Biotype | | | |
|---|---|---|---|---|
| | **RWASA1** | **RWASA2** | **RWASA3** | **RWASA4** |
| **Resistant[a]** | Cltr2401 | Cltr2401 | Cltr2401 | Ripper |
| | Halt | Halt | MTRWA92-161 | Halt |
| | MTRWA92-93 | Ripper | Pan 3144 | ThunderCL |
| | MTRWA92-120 | Ankor | Halt | Cltr2401 |
| | MTRWA92-150 | Pan3144 | MTRWA92-150 | Corwa |
| | MTRWA92-115 | ThunderCL | MTRWA92-149 | MTRWA92-161 |
| | MTRWA92-114 | MTRWA92-91 | MTRWA92-120 | MTRWA92-93 |
| | Ripper | Corwa | MTRWA92-93 | MTRWA92-149 |
| | Pan3144 | MTRWA92-121 | MTRWA92-123 | |
| | MTRWA92-161 | 18FAWWON-SA 62 | MTRWA92-115 | |
| | MTRWA92-91 | MTRWA92-155 | MTRWA92-91 | |
| | MTRWA92-123 | MTRWA92-123 | MTRWA92-121 | |
| | MTRWA92-121 | | MTRWA92-155 | |
| | ThunderCL | | MTRWA92-158 | |
| | MTRWA92-149 | | | |
| | MTRWA92-155 | | | |
| | Gariep | | | |
| | Outlook | | | |
| | MTRWA92-158 | | | |
| **Intermediate[ab]** | Ankor | 18FAWWON-SA 57 | MTRWA92-114 | 18FAWWON-SA 64 |
| | MTRWA92-145 | MTRWA92-114 | Corwa | MTRWA92-155 |
| | PI372129 | Stanton | | Hatcher |
| | Stanton | MTRWA92-161 | | MTRWA92-91 |
| | Corwa | MTRWA92-93 | | |
| | Hatcher | Prowers99 | | |
| | 18FAWWON-SA 57 | 18FAWWON-SA 64 | | |
| | 18FAWWON-SA 64 | MTRWA92-115 | | |
| | Yumar | Gariep | | |
| | BondCL | PI372129 | | |
| | 18FAWWON-SA 62 | | | |
| **Susceptible[b]** | Prowers99 | MTRWA92-120 | Ripper | 18FAWWON-SA 57 |
| | Hugenoot | BondCL | Outlook | Ankor |
| | | MTRWA92-150 | Stanton | MTRWA92-123 |
| | | MTRWA92-158 | BondCL | MTRWA92-114 |
| | | Outlook | MTRWA92-145 | MTRWA92-150 |
| | | MTRWA92-149 | Yumar | Prowers99 |
| | | MTRWA92-145 | 18FAWWON-SA 64 | BondCL |
| | | Yumar | Hatcher | Yumar |
| | | Hatcher | Prowers99 | MTRWA92-115 |
| | | Hugenoot | 18FAWWON-SA 57 | 18FAWWON-SA 62 |
| | | | 18FAWWON-SA 62 | MTRWA92-121 |
| | | | PI372129 | Gariep |
| | | | ThunderCL | Outlook |
| | | | Ankor | PI372129 |
| | | | Hugenoot | Stanton |

*(Continued)*

**Table 3.** (Continued)

| Resistance | Russian Wheat Aphid Biotype | | | |
|---|---|---|---|---|
| | RWASA1 | RWASA2 | RWASA3 | RWASA4 |
| | | | Gariep | MTRWA92-120 |
| | | | | Pan3144 |
| | | | | MTRWA92-145 |
| | | | | MTRWA92-158 |
| | | | | Hugenoot |

stringent Bonferroni correction method. These include marker M545 on chromosome 1A, M5591 on chromosome 6A, M5915 and M6067 on chromosome 6B. These markers showed the highest level of significance in the association study. The majority of the significant markers identified exhibited negative effect sizes, while only 10 MTA had positive effect sizes.

**Linkage disequilibrium.** The LD decayed with increasing physical distances between SNP markers. The average LD throughout the whole genome was 0.38 occurring at distances of approximately 10 Mbp (Fig 3A). The LD among markers with significant association with RWA resistance ranged from very weak ($R^2 < 0.2$) to very strong ($R^2 > 0.8$) correlations spanning a distance of 99.7 centimorgans (cM) (Fig 3B). The LD plot exhibited double peaks of the $R^2$ values found among markers on two LD blocks. The first peak of $R^2$ values above 0.6 was found for markers occurring within the first 20Mbp and the second peak occurred within a 30 Mbp range from 280-310Mbp distance. A single haplotype block consisting of markers M6511, M6087, M5591, M1142 and M6067 was identified (Fig 3B). Seven significant SNP markers (M6511, M6087, M5591, M1142, M4564, M927 and M6067) exhibited linkage disequilibrium ($R^2 > 0.80$, $p < 0.0001$) for RWA resistance.

**Candidate genes and reported quantitative trait loci.** The BLAST analyses using the sequences of significant SNP markers revealed that the MTA overlapped 22 candidate genes with eight of these presently uncharacterised (Table 5). Marker M546 on chromosome 1A

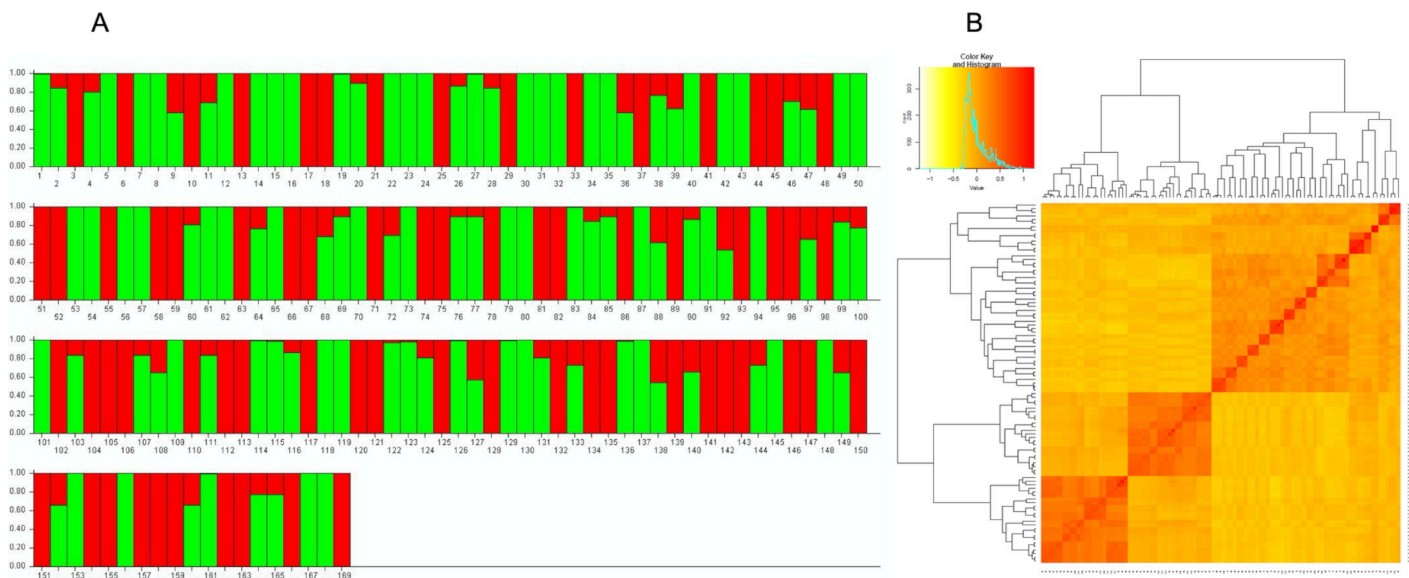

**Fig 1. Population structure of the study panel. A** The two distinct clusters found in 169 wheat bulked samples used in this study. The green bars represent 102 winter-type bulked samples and the red bars represent 67 spring-type bulked samples. **B** Kinship matrix depicting relatedness among the genotypes.

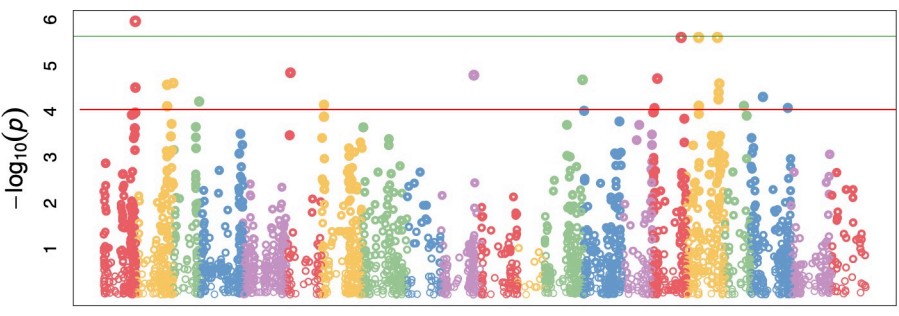

**Fig 2. Manhattan plot showing significant SNP markers associated with RWA resistance using CMLM with a FDR adjusted p-value of 0.05.** The horizontal green and red lines represent the threshold for genome wide significance (p < 0.000001) and FDR adjusted p < 0.0001, respectively.

overlapped the gene *emb2742* while the *PAT1H1* gene was flanked by two markers on chromosome 1B (Table 5). A homologue of the breast cancer susceptibility gene (*BRCA1*) was also identified on chromosome 1B (Table 5). The single marker on chromosome 2D aligned with

**Table 4. Significant SNP markers associated with wheat reaction to Russian wheat aphid biotypes.**

| Marker | Allele | Chr | Position | P-value | MAF | R² | FDR (0.05) | Effect |
|--------|--------|-----|----------|---------|-----|-----|------------|--------|
| M545 | G/A | 1A | 580422454 | 1,09E-06 | 0,12 | 0,47 | 0,0026 | -0,75 |
| M546 | C/T | 1A | 581585515 | 3,03E-05 | 0,46 | 0,44 | 0,0101 | -1,28 |
| M846 | C/T | 1B | 572298712 | 2,63E-05 | 0,09 | 0,44 | 0,0095 | 0,81 |
| M937 | G/A | 1B | 681253335 | 2,42E-05 | 0,07 | 0,44 | 0,0095 | 0,93 |
| M843 | A/G | 1B | 571441403 | 7,75E-05 | 0,07 | 0,43 | 0,0129 | -1,15 |
| M845 | G/A | 1B | 572297017 | 7,75E-05 | 0,43 | 0,43 | 0,0129 | 1,15 |
| M1142 | A/G | 1D | 470730648 | 6,09E-05 | 0,19 | 0,44 | 0,0129 | 0,95 |
| M2366 | A/G | 2D | 76639778 | 1,44E-05 | 0,07 | 0,45 | 0,0095 | 0,99 |
| M2511 | C/T | 3A | 46165651 | 7,17E-05 | 0,25 | 0,44 | 0,0129 | 0,95 |
| M3779 | G/A | 4A | 612113696 | 1,62E-05 | 0,32 | 0,45 | 0,0095 | 1,27 |
| M3780 | G/A | 4A | 612113788 | 1,62E-05 | 0,18 | 0,45 | 0,0095 | -1,27 |
| M4564 | C/T | 5A | 694973102 | 2,05E-05 | 0,19 | 0,45 | 0,0095 | -0,87 |
| M4591 | C/A | 5B | 14510006 | 9,71E-05 | 0,26 | 0,43 | 0,0129 | -0,83 |
| M5390 | T/C | 6A | 32662317 | 8,49E-05 | 0,22 | 0,43 | 0,0129 | -0,87 |
| M5460 | G/A | 6A | 84980487 | 1,93E-05 | 0,28 | 0,45 | 0,0095 | 0,86 |
| M5591 | C/A | 6A | 522610607 | 2,44E-06 | 0,13 | 0,46 | 0,0026 | -1,14 |
| M5915 | A/G | 6B | 227830991 | 2,44E-06 | 0,13 | 0,46 | 0,0026 | -1,14 |
| M5919 | T/C | 6B | 229279621 | 7,59E-05 | 0,11 | 0,43 | 0,0129 | -0,99 |
| M5923 | C/T | 6B | 231349419 | 7,59E-05 | 0,11 | 0,43 | 0,0129 | -0,99 |
| M5928 | G/A | 6B | 232067400 | 7,59E-05 | 0,22 | 0,43 | 0,0129 | -0,50 |
| M6067 | C/T | 6B | 573324217 | 2,44E-06 | 0,13 | 0,46 | 0,0026 | -1,14 |
| M6082 | T/G | 6B | 591507176 | 5,48E-05 | 0,13 | 0,44 | 0,0129 | -0,95 |
| M6087 | A/G | 6B | 595289030 | 3,84E-05 | 0,36 | 0,44 | 0,0119 | -0,95 |
| M6102 | G/A | 6B | 606983489 | 2,48E-05 | 0,36 | 0,44 | 0,0095 | -0,92 |
| M6265 | G/A | 6D | 347684876 | 7,59E-05 | 0,11 | 0,43 | 0,0129 | -0,99 |
| M6511 | T/C | 7A | 223472712 | 4,84E-05 | 0,14 | 0,44 | 0,0129 | -1,47 |
| M6801 | T/C | 7A | 680341517 | 8,41E-05 | 0,35 | 0,43 | 0,0129 | 1,23 |

MAF = minor allele frequency; FDR = false discovery rate.

Chr = chromosome.

**Table 5. Putative candidate genes identified and comparisons with markers for Russian wheat aphid resistance reported in previous studies.**

| Marker | Chromosome | IWGSC gene ID | Gene name | Protein name | Reported SSR marker |
|--------|-----------|---------------|-----------|--------------|---------------------|
| M545 | 1A | TraesCS1A02G425200 | | | Novel |
| M546 | 1A | TraesCS1A02G427100 | emb2742 | EMBRYO DEFECTIVE 2742 | Novel |
| M846 | 1B | TraesCS1B02G343900 | PAT1H1 | Protein PAT1 homolog 1 | Novel |
| M845 | 1B | TraesCS1B02G343900 | PAT1H1 | Protein PAT1 homolog 1 | Novel |
| M843 | 1B | TraesCS1B02G343100 | BRCA1 | Protein BREAST CANCER SUSCEPTIBILITY 1 homolog | Novel |
| M937 | 1B | | | | Novel |
| M1142 | 1D | TraesCS1D02G408300 | | | Novel |
| M2366 | 2D | TraesCS2D02G131200 | RH10 | DEAD-box ATP-dependent RNA helicase 10 | Novel |
| M2511 | 3A | TraesCS3A02G074000 | CSTF77 | Cleavage stimulation factor subunit 77 | Xwmc264 [33] |
| M3779 | 4A | TraesCS4A02G324800 | CPSF73-I | Cleavage and polyadenylation specificity factor subunit 3-I | Novel |
| M3780 | 4A | TraesCS4A02G324800 | CPSF73-I | Cleavage and polyadenylation specificity factor subunit 3-I | Novel |
| M4564 | 5A | TraesCS5A02G538200 | | | Novel |
| M4591 | 5B | | | | Xbarc109 [33] |
| M5390 | 6A | | | | Xgwm1017 [32] |
| M5460 | 6A | | | | Novel |
| M5591 | 6A | TraesCS6A02G291100 | | | Novel |
| M5915 | 6B | TraesCS6B02G193300 | | | Novel |
| M5919 | 6B | TraesCS6B02G194500 | | | Novel |
| M5923 | 6B | TraesCS6B02G195400 | SWI3B | SWI/SNF complex subunit SWI3B | Novel |
| M5928 | 6B | | | | Novel |
| M6082 | 6B | TraesCS6B02G336300 | CHLP | Geranylgeranyl diphosphate reductase, chloroplastic | Novel |
| M6087 | 6B | TraesCS6B02G338100 | | | Novel |
| M6102 | 6B | TraesCS6B02G344600 | | | Novel |
| M6265 | 6D | TraesCS6D02G245500 | | | Novel |
| M6511 | 7A | | | | Novel |
| M6801 | 7A | TraesCS7A02G491300 | UBC22 | Ubiquitin-conjugating enzyme E2 22 | Novel |

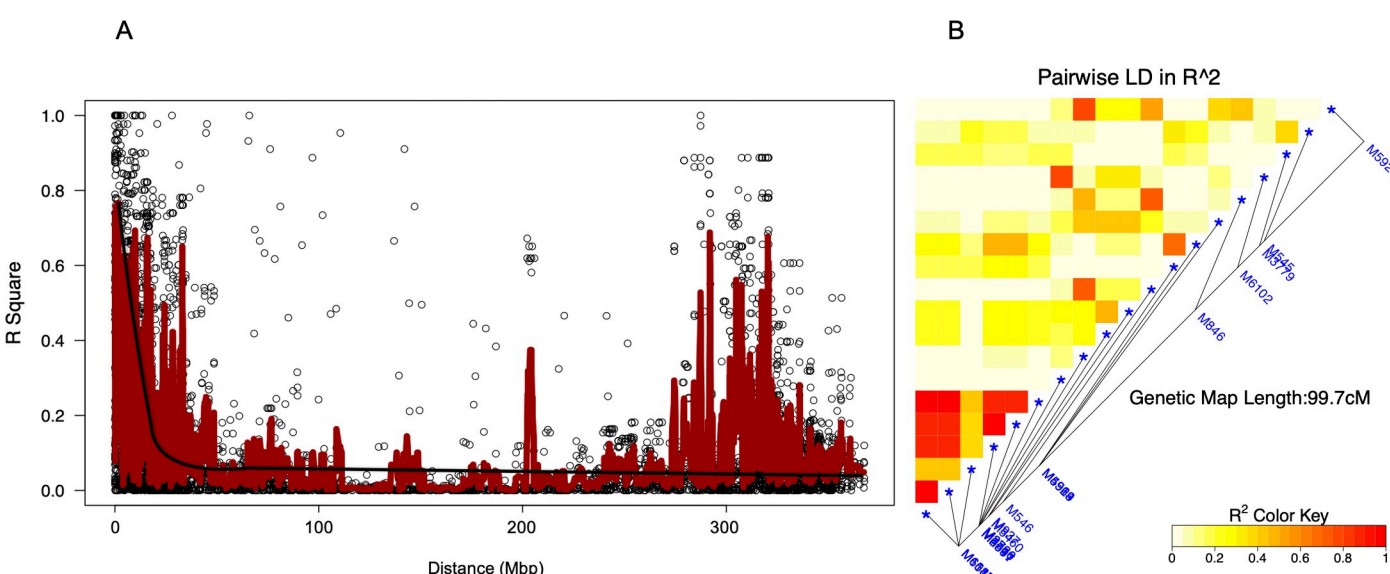

**Fig 3. Linkage disequilibrium (LD) among significant markers. A** Whole-genome LD decay plot against distance (Mbp) with smoothing curve. **B** Local LD among markers in significant association with Russian wheat aphid resistance.

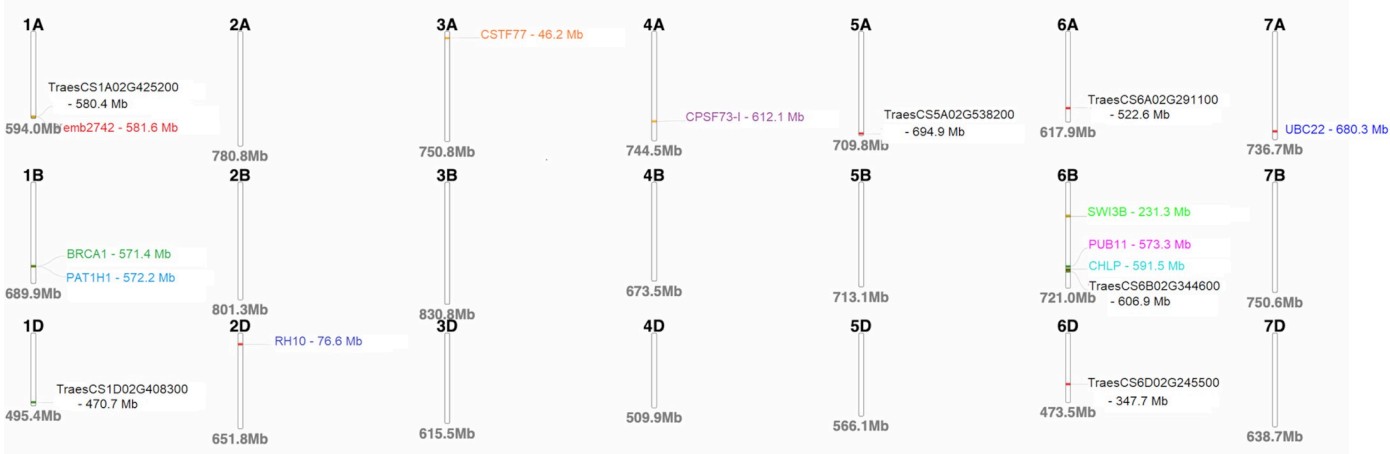

**Fig 4. Physical map of the wheat genome depicting the positions of putative candidate genes identified.**

the *RH10* gene and the gene *CSTF77* was linked to marker M2511 on chromosome 3A (Table 5). Two markers on chromosome 4A aligned to the *CPSF73-I gene* (Table 5). Three candidate genes, namely, *PUB11*, *CHLP* and *SWI3B* were identified on chromosome 6B and the *UBC22* gene was detected on chromosome 7A (Table 5). The functions of these genes encompassed a wide range of biological functions and some are reported to play indirect roles in plant defence pathways. Three markers were linked to previously reported QTLs (Table 5) that included markers M2511, M4591 and M5390 with alignment to SSR markers *Xwmc264*, *Xbarc109* and *Xgwm1017*, respectively. The physical chromosomal location of the genes linked to significant markers are depicted in Fig 4. The physical map showed that several markers were co-localised on chromosomes 1A, 1B and 6B (Fig 4).

## Discussion

Relatively few association mapping studies of RWA resistance have been done to date [41–43]. Prior to that most genetic studies involved linkage mapping of RWA resistance genes. Linkage mapping has relatively low mapping resolution since it is dependent on biparental mapping populations. In contrast, association mapping uses historic recombination events that have accumulated over generations, therefore, greater allelic diversity is obtained with higher resolution [61]. This approach allowed the identification of markers linked to genes with known plant defence involvement.

During an initial SSR marker study of these *Dn4* containing genotypes some intriguing allelic diversity was observed. Several allele sizes were obtained with primer GWM337 (175 bp, 195 bp and 225 bp) that differed from what was reported for these alleles. The allele size (175 bp) obtained from PI372129 correspond to that obtained by Liu et al. [36]. The same 175 bp allele was observed in all the Halt plants tested, however, it conflicted with a 225 bp allele size that was reported by Arzani et al. [35] for *Xgwm337*. The 225 bp allele size was observed for Ankor, Ripper, 18 FAWWON-SA 57, some plants of Corwa and PI586956. The 195 bp allele amplified in susceptible Hugenoot and Yuma by marker *Xgwm337* is consistent with the allele size reported by Liu et al. [36] in the RWA susceptible cultivar Thunderbird. However, this fragment common to susceptible genotypes, was also observed in a number of *Dn4*-containing genotypes, suggesting that allelic homoplasy exists. This was similar to the observation of a 125 bp fragment amplified in PI372129 and Halt and confirmed what was previously reported by

Liu et al. [36] and Arzani et al. [35]. The null-allele for marker *Xgwm106* that was previously associated with susceptibility or *Dn4* absence was present in some expected *Dn4*-carrying germplasm as well as the susceptible controls. This may be due to genotypes containing both the 195 bp (GWM337) and the null allele (GWM106) being linked to a wild type susceptibility allele variant of the *Dn4*-gene or an incomplete, non-resistant allelic form of the *Dn4*-gene.

Marker *Xcfd61* clearly distinguished between *Dn4*-containing genotypes and *Dny*-containing genotypes. The SSR markers beyond *Xcfd61* on the long arm of chromosome 1D need to be investigated further for distinction between *Dn4* and *Dny*. What is of great interest is that alleles that amplified in *Dn4*-containing genotypes were also observed in *Dny*-containing genotypes PI586956, Stanton and 18 FAWWON-SA 57. This poses the question whether these SSR marker alleles are just common non-diagnostic alleles in these bread wheats with associated backgrounds or are they potentially linked to *Dny* as well and if so, whether *Dny* may in fact be an allelic variant of *Dn4* or part of the tightly linked gene cluster on chromosome 1DS.

The results of this first phase of the study indicated that the previously associated and published *Dn4* markers *Xgwm337* and *Xgwm106* may not be diagnostic for the presence of the *Dn4*-gene across all *Dn4* containing germplasm. The diverse SSR marker alleles observed across all *Dn4* genotypes during this study strongly indicate the possibility of a number of different allelic variants of *Dn4* or a tightly linked gene cluster present on chromosome 1DS. This finding lends support to the suggestion by Liu et al. [26] that an allelic gene cluster is present on chromosome 1DS. The results obtained in the SSR marker screening was the premise for conducting the GWAS, aiming to provide clarity on both the phenotypic and allelic variation observed.

The panel of genotypes used in this GWAS exhibited extensive phenotypic variation for RWA resistance, despite the majority of lines reportedly derived from the same resistance source (PI372129—*Dn4* donor). The lines exhibited varying levels of resistance to RWA as previously reported by Tolmay et al. [39]. Since the lines used in this study were not originally developed using South African biotypes, it is likely that they are true breeding only for the trait for which they were selected (resistance to USA biotype RWA1). It is possible that the large phenotypic variation observed may be due to the lines still segregating for alleles or haplotypes associated with resistance specific to the South African biotypes.

The best test analysis revealed five lines with high levels of resistance to four of the South African RWA biotypes namely, Halt, CItr2401, MTRWA92-93, MTRWA92-155 and MTRWA92-161 (Table 3) that is consistent with previous studies [39, 48]. Even though *Dn4* was never used in South African cultivars due to its effectivity failing in other countries, there are still viable sources of *Dn4* containing resistant lines. Thus, selections could be made to purify lines containing effective resistance from the *Dn4* containing lines that may still be economically useful against South African biotypes. Furthermore, broad spectrum resistance to multiple insect pests is desired in cultivars and breeding lines as multiple pests often occur simultaneously in the field. ThunderCL exhibited a high level of resistance to RWASA1, 2 and 4 while Hatcher displayed moderate resistance to RWASA1 and RWASA2. Additionally, these cultivars are reportedly resistant to the Hessian fly (*Mayetiola destructor* Say), whereas they are susceptible to green bug (*Schizaphis graminum* Rondani) [62, 63].

The structure of a mapping population can lead to confounding outcomes in an association analysis if not properly accounted for. Hence, an accurate assessment of the population structure is essential to avoid type I errors [55]. The population structure analysis exhibited a clustering of two distinct subpopulations that is consistent with pedigrees and growth types (Fig 1A). The germplasm was largely divided into winter-type accessions, comprising of cultivars and breeding lines, and spring-type breeding lines. Relatively high levels of differentiation among the clusters were indicated by the mean fixation indices (Cluster 1 = 0.62 and Cluster

2 = 0.31). However, within cluster variation was low as values ranged from 0.17 to 0.21. The presence of admixtures and kinship were expected in this germplasm, since many of the genotypes contain the *Dn4* donor line PI327129 in their pedigrees (Fig 1B; Table 1). Additionally, TAM107 and TAM200 were common parents among several of the genotypes. Cluster 2 which consisted of the spring-type breeding lines share common pedigrees since they are selections from two backcross populations developed in Montana in the United States [64].

Russian wheat aphid resistance in wheat was initially thought to be primarily controlled by single dominant genes located on the 1D and 7D chromosomes. However, recent studies have shown evidence of QTL interactions involved in RWA resistance [21, 32, 33, 40, 65]. Twenty-seven SNP markers were significantly associated with RWA resistance according to the FDR method while four were considered significant with the Bonferroni method. The significant markers were scattered across the wheat genome showing that the markers may not be confined to the D genome only. Negative effective sizes are negatively predictive for a particular phenotype. Since the RWA rating is inversely correlated to resistance, negative effective sizes are indicative of resistance. Therefore 18 of the significant markers identified are considered to be correlated with RWA resistance. Three significant markers (Table 5) were linked with QTL reportedly involved in RWA resistance as previously reported [32, 33]. A marker on chromosome 3A correlated with *Xwmc264* which forms part of a QTL that is associated with the number of expanded leaves [33]. RWA feeding causes the formation of pseudo galls which aids in the protection of the RWA from parasitoids and predators [66]. It also contributes to the plant damage by reducing the photosynthetic area. Leaf expansion may play a role in preventing the formation of the pseudo gall. In addition to being one of the components involved in RWA tolerance, leaf expansion is an important morphological trait that has a direct correlation with yield potential [33]. *Xbarc109* was detected on chromosome 5B and is associated with constituents involved in RWA antibiosis (total fecundity and aphid longevity) [33]. Lastly, marker *Xgwm1017* identified on chromosome 6A is reported as one of the loci involved in an antixenosis response to the Argentinian RWA biotype 2 [32]. Seemingly, the incidence of all categories of RWA resistance mechanisms was significant during the phenotypic evaluation, considering that QTL linked to tolerance, antibiosis and antixenosis overlapped the MTA detected in this study. A single MTA (M1142) was identified on chromosome 1D, which encoded the uncharacterised gene TraesCS1D02G408300. However, the location of this gene differs from the proposed position of the *Dn4* resistance gene. The closest marker that has been reported to the *Dn4* gene is *Xmwg77* [67] which is reportedly located on the short arm of the 1D chromosome at the genetic position of 36.7cM [45]. Markers *Xgwm106* and *Xgwm337* [36], which reportedly flank the *Dn4* gene, are also located on the short arm of 1D at 28 cM and 37.9 cM respectively [45]. M1142 detected in this study is located on the long arm of 1D at the genetic position of 122.8 cM [45], indicating that it is a novel marker. The remaining 23 MTA have not been reported, thus implying that they may be novel alleles involved in RWA resistance. Various studies have reported genetic components involved in RWA resistance on 14 of the 21 wheat chromosomes [7, 9, 16, 18, 19, 21–36]. This study identified new chromosomal regions on 2D, 5A and 6B associated with RWA resistance to four of the South African biotypes. All of the MTA identified on chromosomes 5A and 6B showed a strong correlation with the resistant phenotype whilst the MTA on 2D appears to be correlated with susceptibility. Challenges in characterising significant MTA arose due to the use of multiple marker types and various genetic maps reported by different studies. Consequently, QTL comparisons were made based on approximate genetic distance and should be considered tentative at best. Nevertheless, after an in-depth review of previous association and QTL mapping studies, three of the detected markers overlapped with previously reported QTL on chromosomes 3A, 5B and

6A. Therefore, this GWAS validated that the QTL identified by Castro et al. [32] and Ricciardi et al. [33] are associated with RWA resistance by using different wheat backgrounds.

An overall LD of 0.38 occurring at 10 Mbp was observed (Fig 3A) indicating that the LD decayed at relatively long distance. This could be attributed to narrow genetic variation due to the large proportion of cultivars and elite breeding lines included in the mapping population. This is further compounded by the common geographical region of the majority of the genotypes [43]. A single SNP haplotype marking the 1D, 6A, 6B and 7A QTL was detected, indicating that these markers may be inherited together during a recombination event. Seven significant SNP markers associated with RWA resistance occurred at an average LD of 0.45 over 99.7 cM suggesting moderate linkage. The double peaks in the LD plot could have been caused by the high LD values found among markers on the two LD blocks, one block on chromosome 1 and the other on chromosome 6. The first peak of $R^2$ values above 0.6 was found for markers occurring within the 0–20 Mbp range on chromosome 1 and the second peak occurred within a 30 Mbp range from 280–310 Mbp distance on chromosome 6. Although LD plots would be expected to have a single peak that attenuates with over genetic distance, instances of multiple peaks do occur in structured populations characterised by different haplotypes. Joukhadar et al. [43] found that the double peaks on the LD plot were smoothed by the removal of markers in distant LD blocks that exhibited high $R^2$ values on different chromosomes. Thus, the double peaks in LD found in this study could be attributed to the structure of the germplasm, which was delineated into winter and spring wheat clusters. Each cluster consisted of genotypes with variable reaction to the different aphid biotypes.

Twenty-two putative candidate genes were identified and their inferred biological functions included protein ubiquitination, rRNA processing, mRNA polyadenylation, cell division, embryonic development, chlorophyll biosynthetic processes, antisense RNA processing, chromatin remodelling and DNA repair [68–77]. A large proportion of the candidate genes identified presently remain uncharacterised [68]. Three of these genes may play a role in plant defence responses, through indirect pathways. Marker M843 aligned with the breast cancer susceptibility homologue *BRCA1* originally identified in *Arabidopsis thaliana* [73]. *BRCA1* is an essential gene required for the repair of DNA double strand breaks (DSB) in somatic cells [78] and is a component of DNA damage response (DDR). Evidence suggests that DDR may enhance the activation of plant defence response [79, 80]. While *BRCA1* is a component of the DDR, there is a little evidence to show its involvement in plant defence unlike its variant *BRCA2* [81].

The second putative gene linked to plant defence is *CHLP* that encodes the protein geranylgeranyl diphosphate reductase. The primary function of *CHLP* involves the reduction of geranylgeranyl diphosphate to phytyl diphosphate thereby providing phytol for both chlorophyll and tocopherol synthesis [77, 79]. However, there is evidence that suggests *CHLP* is indirectly involved in plant defence. Tanaka et al. [77] reported that a reduction in geranylgeranyl diphosphate reductase activity induces the loss of chlorophyll and tocopherol. A study by Heng-Moss et al. [82] showed that RWA feeding has an adverse effect on chlorophyll content. Tocopherols have also been shown to regulate the levels of the plants defence hormone jasmonic acid [83] that is known to be regulated upon RWA feeding [84].

*SWI3B* linked with a marker on 6B codes for the protein SWI/SNF complex subunit SWI3B that is a chromatin-remodelling complex involved in the regulation of expression of a considerable number of genes [72]. The interaction between *SWI3B* and abscisic acid-insensitive (ABI) clade proteins may potentially inhibit abscisic acid (ABA) [85]. An inhibition of ABA induced a heightened defence response against *Myzus persicae* [86]. Since *SWI3B* is involved in the suppression of the ABA response, the implication is that the resultant ABA deficiency intensifies defence responses against aphids.

The *Dn4* gene was not detected in this study and this could be due to the possibility of multiple genes required for resistance to different RWA biotypes. Studies have shown that within the same breeding line, certain biotypes require two genes for resistance while others only required one resistance gene [21, 87]. Therefore, it is possible that because the genotyped data was bulked across biotypes prior to association mapping, the presence of *Dn4* was masked by the other genetic components involved in resistance. It is likely that the majority of MTA detected are actually modifiers of the resistance and not the resistance gene itself. However, further investigation is required to confirm this. Research on growth defence trade-offs has become an active focus point over the past few years. Studies have shown that these trade-offs occur based on "decisions" a plant makes in order to sustain its optimal fitness thereby allowing the plant to adjust its phenotypic response to both biotic and abiotic stresses [88]. Chen et al. [89] found that temperatures above 20˚C led to many cultivars showing a susceptible reaction to Hessian wheat fly infestations. Furthermore, Holmes [90] reported that a reduction in light intensity can cause a decrease in sawfly resistance. The effect of temperature [91, 92] and light intensity [93] on the expression of certain resistance genes is also well documented in pathology. This could also be a possibility with RWA; however, this will need to be explored further.

## Conclusion

Whether the *Dn4* resistance is a single dominant gene is questionable, however from the results obtained in this study it is clear that this resistance does not function in isolation. It is either a function of a QTL or it requires the assistance of modifiers. It is well known that RWA resistance is genetically complex and the results obtained in this study could contribute to unravelling the genetic components involved in resistance.

## Supporting information

**S1 Table. Genotypes used in SSR marker screening.**
(XLSX)

**S2 Table. SSR markers used in screening.**
(XLSX)

**S3 Table. RWA damage means.**
(XLSX)

**S4 Table. GAPIT results for RWA resistance.**
(XLSX)

## Acknowledgments

The authors would like to thank Dr Therése Bengtsson and Prof Toi Tsilo for technical assistance.

## Author Contributions

**Conceptualization:** Vicki L. Tolmay, Scott L. Sydenham.

**Data curation:** Lavinia Kisten.

**Formal analysis:** Lavinia Kisten, Isack Mathew, Scott L. Sydenham.

**Funding acquisition:** Vicki L. Tolmay.

**Investigation:** Lavinia Kisten.

**Methodology:** Lavinia Kisten.

**Project administration:** Vicki L. Tolmay, Scott L. Sydenham, Eduard Venter.

**Resources:** Vicki L. Tolmay, Scott L. Sydenham.

**Supervision:** Vicki L. Tolmay, Scott L. Sydenham, Eduard Venter.

**Validation:** Vicki L. Tolmay, Isack Mathew.

**Writing – original draft:** Lavinia Kisten, Scott L. Sydenham.

**Writing – review & editing:** Lavinia Kisten, Vicki L. Tolmay, Isack Mathew, Scott L. Sydenham, Eduard Venter.

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
