## [Decision Letter · Decision Letter 0]

9 Oct 2020

PONE-D-20-27285

Genome-wide association analysis of Russian wheat aphid (*Diuraphis noxia*) resistance in *Dn4* derived wheat lines evaluated in South Africa

PLOS ONE

Dear Dr. Kisten,

Thank you for submitting your manuscript to PLOS ONE. After careful consideration, we feel that it has merit but does not fully meet PLOS ONE’s publication criteria as it currently stands. Therefore, we invite you to submit a revised version of the manuscript that addresses the points raised during the review process.

We look forward to receiving your revised manuscript.

Kind regards,

Aimin Zhang, Ph.D.

Academic Editor

PLOS ONE

Journal Requirements:

2.We note that you are reporting an analysis of a microarray, next-generation sequencing, or deep sequencing data set. PLOS requires that authors comply with field-specific standards for preparation, recording, and deposition of data in repositories appropriate to their field. Please upload these data to a stable, public repository (such as ArrayExpress, Gene Expression Omnibus (GEO), DNA Data Bank of Japan (DDBJ), NCBI GenBank, NCBI Sequence Read Archive, or EMBL Nucleotide Sequence Database (ENA)). In your revised cover letter, please provide the relevant accession numbers that may be used to access these data. For a full list of recommended repositories, see http://journals.plos.org/plosone/s/data-availability#loc-omics or http://journals.plos.org/plosone/s/data-availability#loc-sequencing.

3.Thank you for stating the following in the Financial Disclosure section:

[Funding for the research undertaken in this manuscript were sourced from the following funders:

1.    South African Winter Cereal Trust

https://www.wintercerealtrust.co.za

Grant WCT/W/2018/06 - Awarded to VT

PhD bursary - Awarded to LK

2.    National Research Foundation - Research and Technology Fund

https://www.nrf.ac.za

Grant RTF1505 2911 8466 - Awarded to VT

3.    National Research Foundation - Professional Development Programme

https://www.nrf.ac.za

Block Grant PDP160318160924 - Awarded to VT via Jasper Rees

PhD bursary - Awarded to LK

The funders had no role in study design, data collection and analysis, decision to publish, or preparation of the manuscript.].   

We note that one or more of the authors are employed by a commercial company: LongReach Plant Breeders Management Pty Ltd

Reviewers' comments:

Reviewer's Responses to Questions

**Comments to the Author**

1. Is the manuscript technically sound, and do the data support the conclusions?

Reviewer #1: Yes

Reviewer #2: Yes

2. Has the statistical analysis been performed appropriately and rigorously? 

Reviewer #1: Yes

Reviewer #2: Yes

3. Have the authors made all data underlying the findings in their manuscript fully available?

Reviewer #1: Yes

Reviewer #2: Yes

4. Is the manuscript presented in an intelligible fashion and written in standard English?

Reviewer #1: Yes

Reviewer #2: Yes

5. Review Comments to the Author

Reviewer #1: The manuscript represents a breakthrough contribution to the plant resistance literature for RWA, using an excellent merger of well-known conventional breeding with wheat genomics.

The Manhattan plots of SNP markers associated with RWA resistance completely explodes the idea that RWA resistance genes are dominant traits and reinforces the central theme of the manuscript – that QTLs explain resistance and that QTL-based resistance is strongly influenced by the pedigrees of resistant genotypes.

By combing mapped-based linkage mapping with genomic association mapping results, the authors have demonstrated that an allelic gene cluster is present on chromosome 1DS, but at the same time clearly demonstrated the shortcomings of mapped-based linkage.

The identification of new regions and MTAs on chromosomes 5A and 6B associated with resistance to South African biotypes is a seminal development in the understanding of the complexity of RWA resistance in wheat.

Finally, the identification of plant DR genes with putative function in RWA resistance represents an additional major step in understanding arthropod resistance genes.

At the end of the discussion, the authors may wish to consider adding plant level examples of the effect of temperature and light on expression of resistance i.e. Hessian fly resistance gene sensitivity to temperature (Chen et al. 2014. J. Econ. Entomol. 107:1266) and wheat stem sawfly resistance gene sensitivity to light intensity (Varella et al. 2016 Plant Breed.135: 546).

Minor grammar edits were included in the reviewer attachment.

Reviewer #2: In this study，authors perform a characterization of RWA resistance phenotypes of 32 wheat lines using four South African RWA biotypes and a total of 181 samples were genotyped using the Illumina 9K SNP wheat chip. A GWAS study using 7598 polymorphic SNPs identified 27 marker trait associations (MTA) with an average linkage disequilibrium of 0.38 at 10 Mbp. Twenty putative genes were annotated using the IWGSC RefSeq, three of which are linked to plant defence responses. Technically the strategy seems sound, but I found some processes were not rigorous.

1. Representative samples are required for GWAS analysis. Only 32 samples are used in this paper. The author should elaborate on why these samples are used and how these samples can obtain accurate results of correlation analysis.

2. The relationship between the 32 phenotyped samples and the 181 genotyped samples was not clearly described.

3. Fig. 3A shows that LD has two peaks. Generally, LD is gradually attenuated, what is the reason for forming the double peak of LD?

4. 66

5. Page 16, line 1 an average.

6. Page 5, line 145-147 too long sentence. please split it.

7. Fig. 1 and Fig. 4 is too blurry to be clearly distinguished, please give a higher resolution image.

8. Fig. 21 line 8, ti should be to

6. PLOS authors have the option to publish the peer review history of their article (what does this mean?). If published, this will include your full peer review and any attached files.

Reviewer #1: No

Reviewer #2: No

---

## [Author Response · Author response to Decision Letter 0]

2 Nov 2020

Reviewer 1

Comment

At the end of the discussion, the authors may wish to consider adding plant level examples of the effect of temperature and light on expression of resistance i.e. Hessian fly resistance gene sensitivity to temperature (Chen et al. 2014. J. Econ. Entomol. 107:1266) and wheat stem sawfly resistance gene sensitivity to light intensity (Varella et al. 2016 Plant Breed.135: 546).

Response

The additional examples of the effect temperature and light have on gene expression have been added to the discussion. Page 34 Lines 3-7.

Comment

Minor grammar edits were included in the reviewer attachment.

Response

The grammatically errors were corrected. Throughout document

Reviewer 2

Comment 

Representative samples are required for GWAS analysis. Only 32 samples are used in this paper. The author should elaborate on why these samples are used and how these samples can obtain accurate results of correlation analysis.

Response

The GWAS was based on 181 sampled individuals. These individuals were obtained from 32 genotypes. Within a genotype, there were individuals that exhibited extreme and opposite reactions to a particular RWA biotype. Due to the differential reactions among individuals of the same genotype, it was important to identify the individuals rather than to bulk them as single genotypes. The differences among individuals of same genotypes could arise due to subtle mutations in individuals and crossing over (or recombination) between chromatids of homologous chromosomes during meiosis, which are then inherited by different individual kernels of the same spike. These causes are well known in breeding of self-pollinating species such as wheat where single seed descent is used to counter such variation from influencing breeding progress. Thus, the 181 individuals were deemed sufficient to conduct a GWAS despite their development from 32 genotypes. Page 8 Lines 185-198.

Comment

The relationship between the 32 phenotyped samples and the 181 genotyped samples was not clearly described.

Response

The test individuals were developed based on the response of the thirty-two genotypes to four RWA biotypes. Each genotype was planted and inoculated with four different RWA biotypes in three replicates of 5 plants each. After inoculation and disease severity scoring, plants of the same genotype in the same replicate that exhibited differential response to RWA were bulked separately. A genotype could potentially have individual plants that exhibited susceptibility, intermediate and resistance to RWA. The individuals in the intermediate reaction grouped were not sampled. Only individuals with extreme ie either susceptible or resistant reaction were sampled and bulked accordingly per biotype. Some genotypes consisted of individuals with similar reaction to a particular RWA biotype, in which case a single bulk sampled would be collected. Other genotypes consisted of individuals with differential reaction and at most, two bulk samples of susceptible and resistant samples would be collected per biotype. In total, 181 individuals with differential reaction to the four RWA biotypes were sampled for genotyping and used for downstream analyses. Page 8 Lines 185-198.

Comment

Fig. 3A shows that LD has two peaks. Generally, LD is gradually attenuated, what is the reason for forming the double peak of LD?

Response 

The double peaks in the LD plot could have been caused by the high LD values found among markers on the two LD blocks, one block on chromosome 1 and the other on chromosome 6. The first peak of R2 values above 0.6 was found for markers occurring within the 0-20 Mbp range on chromosome 1 and the second peak occurred within a 30 Mbp range from 280-310 Mbp distance on chromosome 6. Although LD plots would be expected to have a single peak that attenuates with over genetic distance, instances of multiple peaks do occur in structured populations characterised by different haplotypes. Joukhadar et al 2013 found that the double peaks on the LD plot were smoothed by the removal of markers in distant LD blocks that exhibited high R2 values on different chromosomes. Thus, the double peaks in LD found in this study could be attributed to the structure of the germplasm, which was delineated into winter and spring wheat clusters. Each cluster consisted of genotypes with variable reaction to the different aphid biotypes. Page 22 Lines 9-12; Page 31 Lines 18-25; Page 32 Lines 1-5.

Comment

66

Response

It is unclear to what this comment is referring to, therefore it was not addressed.

Comment

Page 16, line 1 an average.

Response 

The error has been corrected. Page 20 Line 9.

Comment 

Page 5, line 145-147 too long sentence. please split it.

Response 

The sentence has been simplified. Page 7 Line 150.

Comment 

Fig. 1 and Fig. 4 is too blurry to be clearly distinguished, please give a higher resolution image.

Response 

The quality of the figures has been improved and higher resolution images were submitted. 

Comment 

Fig. 21 line 8, ti should be to

Response 

The error has been corrected. Page 26 Line 8.

---

## [Decision Letter · Decision Letter 1]

27 Nov 2020

PONE-D-20-27285R1

Genome-wide association analysis of Russian wheat aphid (*Diuraphis noxia*) resistance in *Dn4* derived wheat lines evaluated in South Africa

PLOS ONE

Dear Dr. Kisten,

Thank you for submitting your manuscript to PLOS ONE. After careful consideration, we feel that it has merit but does not fully meet PLOS ONE’s publication criteria as it currently stands. Therefore, we invite you to submit a revised version of the manuscript that addresses the points raised during the review process.

We look forward to receiving your revised manuscript.

Kind regards,

Aimin Zhang, Ph.D.

Academic Editor

PLOS ONE

Reviewers' comments:

Reviewer's Responses to Questions

**Comments to the Author**

1. If the authors have adequately addressed your comments raised in a previous round of review and you feel that this manuscript is now acceptable for publication, you may indicate that here to bypass the “Comments to the Author” section, enter your conflict of interest statement in the “Confidential to Editor” section, and submit your "Accept" recommendation.

Reviewer #3: (No Response)

Reviewer #4: All comments have been addressed

2. Is the manuscript technically sound, and do the data support the conclusions?

Reviewer #3: Yes

Reviewer #4: Yes

3. Has the statistical analysis been performed appropriately and rigorously? 

Reviewer #3: Yes

Reviewer #4: Yes

4. Have the authors made all data underlying the findings in their manuscript fully available?

Reviewer #3: Yes

Reviewer #4: Yes

5. Is the manuscript presented in an intelligible fashion and written in standard English?

Reviewer #3: Yes

Reviewer #4: Yes

6. Review Comments to the Author

Reviewer #3: Dear authors,

I have some suggestions or corrections in the revised manuscript.

1. In page 6, line 137-139, Table 1 was stated that there are 25 genotypes containing Dn4 genes. However, there are 24 genotypes carrying the Dn4 gene in table 1. Also, the sum of the specified numbers of genotypes makes 33 in the same lines.

2. In page 14, line 4-5, the sentence

“Three of the MTRWA92 breeding lines (91, 120 and 160), contained the 195 bp fragment, while the remaining nine lines (93, 114, 115, 121, 145, 149, 150, 155, 158) had the 175 bp fragment.”

MTRWA92-160 line does not contain the 195 bp fragment in Table 2.

3. In page 14, line 11-12, the sentence

“The 125 bp fragment was amplified in PI372129, Halt, 12 BondCL, Yumar and all of the 18 FAWWON-SA 64 plants. ”

Also, Corwa genotype contains the 125 bp fragment according to Table2.

4. In page 18, line 10-11,

“MTRWA92-161 was resistant to RWASA1, RWASA3 and RWASA4 and showed a moderate resistance to RWASA2.”

This information was given two sentences before in the same paragraph, in that;

“Genotypes MTRWA92- 161 and MTRWA92-93 were moderately resistant to RWASA2 and showed a high level of resistance to RWASA1, RWASA3 and RWASA4.”

Best regards

Reviewer #4: It is suggested that some description in the discussion for the resistance level to of the test wheat lines to the other wheat aphids, because the several wheat aphids species usually concur at the same.

7. PLOS authors have the option to publish the peer review history of their article (what does this mean?). If published, this will include your full peer review and any attached files.

Reviewer #3: No

Reviewer #4: No

---

## [Author Response · Author response to Decision Letter 1]

2 Dec 2020

Reviewer 3

Comment

In page 6, line 137-139, Table 1 was stated that there are 25 genotypes containing Dn4 genes. However, there are 24 genotypes carrying the Dn4 gene in table 1. Also, the sum of the specified numbers of genotypes makes 33 in the same lines.

Response

The error has been corrected. Page 6 Line 137.

Comment

In page 14, line 4-5, the sentence

“Three of the MTRWA92 breeding lines (91, 120 and 160), contained the 195 bp fragment, while the remaining nine lines (93, 114, 115, 121, 145, 149, 150, 155, 158) had the 175 bp fragment.”

MTRWA92-160 line does not contain the 195 bp fragment in Table 2.

Response

The error has been corrected within the paragraph. Page 14 Lines 4-5.

Comment

In page 14, line 11-12, the sentence

“The 125 bp fragment was amplified in PI372129, Halt, 12 BondCL, Yumar and all of the 18 FAWWON-SA 64 plants.”

Also, Corwa genotype contains the 125 bp fragment according to Table2.

Response

Corwa has been added to the list of genotypes that contain the 125 bp fragment. Page 14 Line 12.

Comment

In page 18, line 10-11,

“MTRWA92-161 was resistant to RWASA1, RWASA3 and RWASA4 and showed a moderate resistance to RWASA2.”

This information was given two sentences before in the same paragraph, in that;

“Genotypes MTRWA92- 161 and MTRWA92-93 were moderately resistant to RWASA2 and showed a high level of resistance to RWASA1, RWASA3 and RWASA4

Response

The duplicated description of the result has been removed Page 18 Line 10.

Reviewer 4

Comment 

It is suggested that some description in the discussion for the resistance level to of the test wheat lines to the other wheat aphids, because the several wheat aphids species usually concur at the same.

Response

The resistance response of the genotypes to other aphid species was not assessed in this study. Our focus was on the Russian aphid, although the other species of aphids may occur simultaneously with the Russian aphid. The reaction of the genotypes to other aphid species could not be immediately established from literature. However, reaction of two of the genotypes, namely Hatcher and ThunderCL, to other insect pests, namely Hessian fly and green bug were included in the discussion as follows: 

“Furthermore, broad spectrum resistance to multiple insect pests is desired in cultivars and breeding lines as multiple pests often occur simultaneously in the field. ThunderCL exhibited a high level of resistance to RWASA1, 2 and4 while Hatcher displayed moderate resistance to RWASA1 and RWASA2. Additionally, these cultivars are reportedly resistant to the Hessian fly (Mayetiola destructor Say), whereas they are susceptible to green bug (Schizaphis graminum Rondani) [62,63].” Page 28 Lines 18-23.

---

## [Decision Letter · Decision Letter 2]

10 Dec 2020

Genome-wide association analysis of Russian wheat aphid (*Diuraphis noxia*) resistance in *Dn4* derived wheat lines evaluated in South Africa

PONE-D-20-27285R2

Dear Dr. Kisten,

We’re pleased to inform you that your manuscript has been judged scientifically suitable for publication and will be formally accepted for publication once it meets all outstanding technical requirements.

Kind regards,

Aimin Zhang, Ph.D.

Academic Editor

PLOS ONE

Additional Editor Comments (optional):

Reviewers' comments:

Reviewer's Responses to Questions

**Comments to the Author**

1. If the authors have adequately addressed your comments raised in a previous round of review and you feel that this manuscript is now acceptable for publication, you may indicate that here to bypass the “Comments to the Author” section, enter your conflict of interest statement in the “Confidential to Editor” section, and submit your "Accept" recommendation.

Reviewer #3: (No Response)

2. Is the manuscript technically sound, and do the data support the conclusions?

Reviewer #3: (No Response)

3. Has the statistical analysis been performed appropriately and rigorously? 

Reviewer #3: (No Response)

4. Have the authors made all data underlying the findings in their manuscript fully available?

Reviewer #3: (No Response)

5. Is the manuscript presented in an intelligible fashion and written in standard English?

Reviewer #3: (No Response)

6. Review Comments to the Author

Reviewer #3: (No Response)

7. PLOS authors have the option to publish the peer review history of their article (what does this mean?). If published, this will include your full peer review and any attached files.

Reviewer #3: No

---

## [Editor Report · Acceptance letter]

15 Dec 2020

PONE-D-20-27285R2 

Genome-wide association analysis of Russian wheat aphid *(Diuraphis noxia)* resistance in Dn4 derived wheat lines evaluated in South Africa 

Dear Dr. Kisten:

I'm pleased to inform you that your manuscript has been deemed suitable for publication in PLOS ONE. Congratulations! Your manuscript is now with our production department. 

Kind regards, 

on behalf of

Prof. Aimin Zhang 

Academic Editor

PLOS ONE